# Childhood Trauma, Coping Styles, and Mental Health in Polycystic Ovarian Syndrome

**DOI:** 10.3390/bs15111551

**Published:** 2025-11-14

**Authors:** Aimee N. Schneider, Brian N. Chin

**Affiliations:** Department of Psychology, Trinity College, Hartford, CT 06106, USA; aimee.schneider@trincoll.edu

**Keywords:** polycystic ovarian syndrome, childhood trauma, coping styles, mental health, disordered eating, quality of life, mediation analysis, behavioral health

## Abstract

Polycystic ovarian syndrome (PCOS) is a common but understudied endocrine condition associated with elevated mental health risks, including disordered eating and reduced quality of life. Prior research suggests that childhood trauma may contribute to these outcomes, yet the psychological mechanisms underlying this link remain poorly understood. In this study, we examined whether coping styles account for the relationship between childhood trauma and mental health outcomes among individuals with PCOS. Participants (*N* = 150) completed validated measures of childhood trauma, disordered eating symptoms, PCOS-related quality of life, and coping behaviors. As expected, greater trauma exposure was associated with more disordered eating and poorer quality of life. Analyses indicated these associations were partially accounted for by higher disengagement coping. These findings identify coping styles as a candidate mechanism linking early-life adversity to mental health outcomes in PCOS, suggesting targets for intervention and screening.

## 1. Introduction

### 1.1. Literature Review

Polycystic ovarian syndrome (PCOS) is a common endocrine condition that affects an estimated 8–13% of individuals of reproductive age ([29]). Despite its prevalence, PCOS remains both under-researched and insufficiently recognized in the clinical and psychological literature. Long diagnostic delays are common—with nearly one in three individuals waiting over two years and consulting three or more providers before receiving a diagnosis—and many report feeling dissatisfied with the information received from their providers ([11]). Recent electronic health record analyses have further highlighted these systemic gaps, showing that a large proportion of individuals meeting PCOS criteria still receive no formal diagnosis, especially those who are racial/ethnic minorities or publicly insured ([24]). This pattern reflects broader structural inequities in women’s health research; indeed, most clinical trials excluded women altogether until the 1990s, with enduring consequences for diagnostic accuracy ([23]). These diagnostic delays and inequities inform the present study’s focus on examining behavioral pathways that may help account for the mental health burden in PCOS.

Although PCOS is clinically defined by hormonal and reproductive symptoms, it is also associated with substantial yet often overlooked mental health challenges. Individuals with PCOS are more likely to report depression, anxiety, disordered eating, and reduced quality of life ([5]; [6]; [17]). These mental health risks may reflect the cumulative toll of chronic symptoms, weight-centered treatment approaches, and the emotional strain of navigating a condition that is frequently misunderstood or minimized in healthcare settings ([6]; [11]). However, less is known about the psychological and behavioral pathways through which these outcomes emerge. This knowledge gap highlights the need to understand how past experiences shape current mental health in individuals with PCOS.

Childhood trauma—including experiences of abuse, neglect, or chronic stress—has been linked with a range of negative long-term health outcomes, including depression, anxiety, chronic illness, and disability ([9]; [18]). While this association is well-established in the general population, less research has explored its relevance to individuals with PCOS. A small but growing body of literature suggests that people with PCOS may report higher rates of early childhood adversity than those without PCOS. For example, [25] ([25]) found that Australian women with PCOS reported more adverse childhood experiences than women without PCOS, and [22] ([22]) observed similar patterns in a South African sample, where emotional abuse emerged as the strongest predictor of PCOS status when controlling for other types of maltreatment. Childhood trauma predicts poorer mental health across populations, not only among individuals with PCOS. We focus on PCOS here because condition-specific stressors, stigma, and care barriers may exacerbate trauma-related vulnerabilities.

These findings suggest that early adversity may be one factor contributing to the mental health burden observed in PCOS populations. However, most existing studies have focused on prevalence comparisons rather than examining explanatory mechanisms, and to our knowledge, no published studies have examined this association in a U.S.-based sample. Replication in U.S. cohorts is especially important, as most existing findings come from Australia and South Africa ([22]; [25]), and may not generalize to U.S. clinical or cultural contexts. Consequently, the potential pathways linking childhood trauma to adult mental health in PCOS remain insufficiently explored. Identifying the processes that help explain how early life adversity contributes to current mental health outcomes is important for identifying modifiable targets for intervention.

One candidate mechanism underlying this association is coping styles, the strategies that individuals use to manage stressful situations, which vary in their adaptiveness and psychological consequences. Contemporary models of stress and coping distinguish between engagement coping—which involves approaching or actively managing stressors—and disengagement coping—which involves avoiding, withdrawing, or emotionally numbing ([4]). These dimensions are especially relevant in health-related contexts where disengaged coping has been linked with worse psychological outcomes, including depression, anxiety, disordered eating, and reduced quality of life ([2]; [8]). Given that childhood trauma can influence how individuals cope with stress later in life, coping styles represent a plausible behavioral mechanism through which early life adversity may shape current mental health outcomes in PCOS.

### 1.2. Present Study and Hypotheses

The present study builds on prior research linking early life adversity and psychological distress in PCOS by examining whether coping styles help account for this association. Specifically, we tested whether childhood trauma was associated with two mental health outcomes: disordered eating symptoms and PCOS-related quality of life. We hypothesized that greater childhood trauma exposure would predict more disordered eating symptoms and poorer quality of life. We also conducted exploratory mediation analyses to assess whether coping styles—particularly engagement and disengagement coping—helped to account for these associations. Given the cross-sectional design, these models were intended to be hypothesis-generating rather than confirmatory. Analyses controlled for age, race, and educational attainment based on prior links with coping and health disparities in PCOS and related populations.

## 2. Materials and Methods

### 2.1. Participants and Procedures

This cross-sectional survey study was approved by the Trinity College Institutional Review Board (Protocol ID: 4923) and conducted in accordance with relevant ethical guidelines, including the Declaration of Helsinki. Eligible participants were at least 18 years old, fluent in English, and had received a formal PCOS diagnosis from a medical provider. Data were collected between 19 January 2025 and 27 February 2025.

Participants were recruited through targeted posts on Reddit forums focused on PCOS and women’s health (e.g., r/PCOSRECIPES, r/LeanPCOS, r/PCOS_management, r/WomensHealth, etc.), as well as through physical fliers featuring a QR code that linked to the survey. Online recruitment was used to enable geographically diverse data collection and because access to clinic populations was limited. After providing electronic informed consent, participants completed a questionnaire administered via Qualtrics. Upon completion, participants were given the option to enter a separate, unlinked survey for a raffle to win one of six 25 USD Amazon gift cards; the gift card raffle was funded by departmental research support.

### 2.2. Measures

Participants reported their age, sex assigned at birth, current gender identity, race/ethnicity, educational attainment, country of residence, and perceived socioeconomic status during childhood. Participants were also asked to self-report whether they had received a formal PCOS diagnosis from a medical provider; only those who answered “yes” were eligible to complete the full survey. All instruments were chosen for their established reliability and prior use in adult behavioral health and women’s health research contexts.

Childhood trauma was assessed using the Childhood Trauma Questionnaire—Short Form ([3]), a 28-item self-report measure that assesses five domains of maltreatment: emotional abuse, physical abuse, sexual abuse, emotional neglect, and physical neglect. Items are rated on a five-point Likert scale from 1 (never true) to 5 (very often true), with higher scores indicating greater childhood trauma severity. This scale demonstrated strong internal consistency in our sample (α = 0.89).

Coping style was assessed using the Coping Strategies Inventory—Short Form ([1]), a 16-item self-report measure that assesses two types of coping strategies: disengagement-focused and engagement-focused. Participants rated how frequently they used each strategy on a five-point Likert scale from 1 (never) to 5 (always). Subscale scores were computed for engagement coping (α = 0.71) and disengagement coping (α = 0.76). Higher scores indicated greater use of that coping style. This measure assesses current coping tendencies rather than childhood coping.

Disordered eating was assessed using the Eating Disorder Examination Questionnaire ([10]), a 31-item self-report measure that assesses four domains of eating-related attitudes and behaviors over the past 28 days: eating concern, restraint, shape concern, weight concern). Items are rated on a seven-point Likert scale. Due to minor programming errors, the fifth anchor point on the seven-point scale was inadvertently omitted for the first 12 items and Item 21 was rated on a four-point scale (not at all, slightly, moderately, markedly). Responses were recoded to reflect a 1–7 distribution, consistent with the original scale, and then averaged to compute a total score, with higher scores indicating more disordered eating behaviors. This scale demonstrated excellent internal consistency in our sample (α = 0.93).

PCOS-related quality of life was assessed using the Polycystic Ovary Syndrome Health-Related Quality-of-Life Questionnaire ([20]), a 43-item self-report measure that assesses the psychosocial, reproductive, and physical impacts of PCOS across six domains: body image, emotional distress, fertility concerns, hirsutism, obesity and menstrual symptoms, and sexual function. Items are rated on a seven-point Likert scale from 1 (not at all) to 7 (extremely), with higher scores indicating poorer quality of life. Two items were modified for clarity or inclusivity (e.g., “fear of abortion” was changed to “fear of miscarriage”). Two items concerning lack of sexual desire and dissatisfaction in the role of wife were excluded. Responses were averaged to compute a total score, with higher scores indicating poorer quality of life. The modified 41-item scale demonstrated excellent internal consistency in our sample (α = 0.94).

### 2.3. Data Analysis

Analyses were conducted using SPSS Statistics, Version 29.0 (IBM Corp., Armonk, NY, USA). Prior to conducting formal hypothesis testing, we assessed the internal consistency of all multi-item scales used in the study and verified model assumptions using checks of normality and homoscedasticity.

To test our primary hypotheses, we conducted two separate linear regressions using childhood trauma scores to predict (1) disordered eating symptoms and (2) PCOS-related quality of life. Each regression model included covariates for age, race, and educational attainment. These variables were included due to their known associations with health disparities and coping styles.

In addition, we conducted exploratory mediation analyses using Model 4 of the PROCESS macro ([13]) to assess whether coping styles (disengagement, engagement) accounted for the associations between childhood trauma and the two mental health outcomes. Indirect effects were estimated using 5000 bias-corrected bootstrap samples with 95% confidence intervals. Given the cross-sectional design, mediation models were considered hypothesis-generating rather than confirmatory. Given multiple models and exploratory mediation, we interpret effects with attention to consistency and effect sizes, acknowledging elevated Type I error risk.

### 2.4. Transparency and Openness

This study was not preregistered. The anonymized data and analysis syntax will be made available upon publication at https://osf.io/bqw9g/ (accessed on: 31 July 2025).

## 3. Results

### 3.1. Descriptive Statistics

The final analytic sample included 150 participants who reported a medical diagnosis of PCOS. All participants were assigned female at birth, and the majority identified as women (92.7%) with fewer identifying as non-binary or another gender (6.7%) or men (0.7%). Participants were diverse in age—48.7% were between 25 and 34, 26.7% were 18–24, 20.7% were 35–44, and 3.9% were older than 45. Most participants identified as White (80%) with fewer identifying as Hispanic or Latino/a/x (9%), Asian (8%), Native Hawaiian or Other Pacific Islander (7%), Middle Eastern or North African (2%), Black or African American (2%), or another race (7%). Approximately two-thirds of participants had completed at least a bachelor’s degree and described their childhood socioeconomic status as working or middle class. Participants resided across six continents; most lived in the United States (63%). Scale descriptives are presented in Table 1.

### 3.2. Associations of Childhood Trauma and Mental Health Outcomes

We tested our primary hypotheses by conducting separate linear regressions using childhood trauma scores and covariates to predict disordered eating symptoms and PCOS-related quality of life.

In the model predicting disordered eating, only childhood trauma (B = 0.03, SE = 0.01, β = 0.39, *p* < 0.001, 95% CI [0.02, 0.04]) was a significant predictor. Age, race, and education were included as covariates but were not significant predictors (*p*s > 0.24).

In the model predicting poorer quality of life, childhood trauma (B = 0.62, SE = 0.18, β = 0.29, *p* < 0.001, 95% CI [0.27, 0.98]) and age (B = −8.52, SE = 4.25, β = −0.16, *p* = 0.047, 95% CI [−16.92, −0.11]) were significant predictors. Race and education were included as covariates but were not significant predictors (*p*s > 0.54).

### 3.3. Exploratory Analyses of Coping Styles as Indirect Pathways

We next conducted exploratory mediation analyses to examine whether coping styles (engagement and disengagement) accounted for the associations between childhood trauma and mental health outcomes, controlling for age, race, and education. Conceptual models and key coefficients for each mediation pathway are shown in Figure 1 for disordered eating and Figure 2 for PCOS-related quality of life.

In the model predicting disordered eating, childhood trauma was significantly associated with more disengagement coping (B = 0.01, SE = 0.00, β = 0.29, *p* = 0.001) and less engagement coping (B = −0.01, SE = 0.00, β = −0.40, *p* < 0.001). In turn, disengagement coping (B = 0.59, SE = 0.19, β = 0.28, *p* = 0.002) but not engagement coping (B = −0.18, SE = 0.23, β = −0.08, *p* = 0.43) was significantly associated with more disordered eating symptoms. The completely standardized indirect effect of disengagement coping was significant (β = 0.08, SE = 0.04, 95% CI [0.02, 0.17]) while the completely standardized indirect effect of engagement coping was not significant (β = 0.03, SE = 0.04, 95% CI [−0.04, 0.11]). The direct effect of trauma remained significant (B = 0.02, SE = 0.01, β = 0.28, *p* = 0.002), indicating a small indirect effect through disengagement coping.

In the model predicting poorer quality of life, childhood trauma was again significantly associated with more disengagement coping (B = 0.01, SE = 0.00, β = 0.29, *p* = 0.001) and less engagement coping (B = −0.01, SE = 0.00, β = −0.40, *p* < 0.001). In turn, disengagement coping (B = 23.87, SE = 5.55, β = 0.37, *p* < 0.001) but not engagement coping (B = −7.94, SE = 6.70, β = −0.11, *p* = 0.24) was significantly associated with poorer quality of life. The completely standardized indirect effect of disengagement coping was significant (β = 0.11, SE = 0.04, 95% CI [0.04, 0.19]) while the completely standardized indirect effect of engagement coping was not significant (β = 0.04, SE = 0.04, 95% CI [−0.03, 0.12]). The direct effect of trauma remained significant (B = 0.40, SE = 0.18, β = 0.18, *p* = 0.030), indicating a small indirect effect through disengagement coping.

## 4. Discussion

This study examined whether childhood trauma was associated with mental health outcomes among individuals with PCOS and whether coping styles helped account for those associations. Consistent with our hypotheses, greater trauma exposure was associated with more disordered eating symptoms and poorer PCOS-related quality of life. Exploratory analyses indicated that these effects were partially accounted for by disengagement coping, the tendency to rely on stress management strategies characterized by physically or emotionally withdrawing from the stressor rather than actively addressing it. In contrast, engagement coping did not show a significant indirect effect. These exploratory findings suggest that how individuals respond to stress may help to account for why early adversity confers psychological risk in the context of PCOS.

The association of childhood trauma and psychological distress is well-documented in general populations, including consistent links with depression, anxiety, and disordered eating ([15]; [18]). Our findings build on this literature by showing that childhood trauma is also linked with poorer mental health outcomes in a PCOS context, addressing a gap in prior research, which has emphasized prevalence and symptom profiles over underlying behavioral mechanisms.

The current results suggest that disengagement coping may play a small but statistically significant role in linking early adversity to current mental health outcomes. The magnitude of the indirect effects was small, and interpretations are correspondingly conservative. This finding is consistent with theoretical models of coping that describe disengagement strategies—such as denial, emotional numbing, self-isolation, or behavioral withdrawal—as maladaptive in the context of chronic health stressors ([2]; [4]). While engagement coping was negatively associated with trauma, its impact did not explain additional variance in disordered eating or quality of life. These results suggest that the mental health challenges experienced by individuals with PCOS may reflect, in part, habitual coping patterns shaped by early adversity. In turn, these patterns may hinder effective regulation of stress and other emotions and reduce engagement with support ([14]; [19]).

These behavioral findings also align with neuroendocrine theories of allostatic load and biological embedding, which suggest that chronic early-life stress can disrupt hormonal regulation through chronic activation of the HPA axis and related systems ([7]; [19]). In theory, this may contribute to the hormonal dysregulation and metabolic burden observed in PCOS populations, representing a biologically plausible pathway through which trauma history and coping styles can influence long-term health outcomes. Consistent with this account, chronic activation of the HPA axis during sensitive developmental periods has been linked with both elevated and suppressed cortisol levels—markers of hormonal dysregulation ([12]). These disruptions are consistent with the higher rates of insulin resistance and metabolic dysregulation observed in individuals with PCOS, especially among those with trauma histories ([7]). In addition to their metabolic consequences, childhood trauma and HPA-axis dysregulation may also contribute to sleep disruption which, in turn, can amplify emotional reactivity ([21]), impair coping ([28]), and even further dysregulate endocrine and metabolic systems ([27]).

Weight-related stigma and pressure to lose weight are also common in PCOS care, and may exacerbate psychological distress by reinforcing shame and avoidance-based coping ([16]). Prior studies have documented elevated rates of disordered eating and reduced quality of life among individuals with PCOS ([5]; [17]), highlighting the need for interventions that address both patients’ coping responses and the environmental stressors that elicit and shape them. Building on these findings, our results suggest that reducing reliance on avoidance-based coping—via interventions like behavioral activation, emotion regulation training, or trauma-informed psychoeducation—may be a promising pathway for improving psychological outcomes. Because these coping patterns often emerge as self-protective adaptations to early adversity, patients may not recognize their long-term costs. Brief, structured conversations about coping during routine care could help normalize these responses and identify individuals who may benefit from additional support.

These findings may have practical relevance for clinicians working with PCOS populations in primary care, OB/GYN, or behavioral health settings. Routine screening for childhood trauma may help identify patients who have an elevated risk for poor mental health outcomes. Understanding that early life adversity can shape stress responses and coping styles may help inform more personalized care. Clinicians might also consider discussing coping strategies as part of routine care. Even brief conversations about how patients typically respond to stress may help to identify opportunities for more targeted support. This is especially important given the stigma, frustration, and emotional burden reported by individuals navigating PCOS in clinical settings ([11]). Both brief psychological screenings and referrals for behavioral support resources are consistent with international guidelines for PCOS care ([26]) and may help reduce barriers to treatment for those with unmet psychosocial needs.

This study has several limitations. First, its cross-sectional design precludes causal inference. Although the mediation models were theoretically based and statistically supported, longitudinal research is needed to clarify the direction and temporal precedence of these associations. Second, these data were self-reported and may be influenced by recall bias. While this concern is partially mitigated by using previously validated instruments, future research would benefit from incorporating multiple measurement sources, including clinician ratings or biometric assessments. Relatedly, PCOS status also was self-reported, which may introduce diagnostic misclassification; any nondifferential error would likely attenuate associations. Third, the sample was geographically diverse but skewed toward Western, educated participants with internet access and Reddit familiarity. Recruitment via Reddit likely skewed toward younger, digitally engaged participants, and the sample was predominantly White, which limits generalizability. Fourth, a minor programming error altered response options for a subset of EDE-Q items; reliability remained high, but comparability to unmodified administrations may be limited. Similarly, modifications to the quality of life questionnaire may have slightly altered construct coverage compared to the original instrument and limit comparability to earlier studies using the unmodified version. Fifth, we estimated several models which may have increased Type I error risk. Finally, the measure used to assess coping was validated but not tailored to PCOS-specific stressors. Future work should develop and validate PCOS-adapted coping instruments to improve construct specificity.

Future research should investigate how coping styles evolve over time, and whether trauma-informed interventions can promote more adaptive coping responses among individuals with PCOS. Longitudinal studies are needed to determine whether disengagement coping contributes to the development of poorer mental health or arises in response to stress and mental health challenges. Intervention studies should also explore whether brief, skill-based programs, such as emotion regulation training or mindfulness interventions, may help to buffer the psychological impacts of early adversity in this population.

Researchers should also prioritize expanding recruitment to reach more racially, culturally, and socioeconomically diverse samples. Given the documented disparities in PCOS diagnosis and care, it is critical that future work include populations historically underrepresented in research ([24]). Future research could also explore how sleep disruption interacts with coping and childhood trauma exposure in shaping mental health outcomes among individuals with PCOS. Given that sleep disruption is linked with emotional dysregulation ([21]) and metabolic dysfunction ([27]), sleep may represent an important and potentially modifiable factor in this context. Continued development of trauma-informed care may help bridge the gap between patients’ lived experiences and current clinical management of PCOS.

## 5. Conclusions

This study contributes to a growing understanding of how psychological and behavioral factors influence health and well-being in individuals with PCOS. Our findings suggest that childhood trauma is associated with more disordered eating symptoms and poorer quality of life, and that these associations are partially accounted for by disengagement coping. Supporting patients in developing more adaptive coping strategies, particularly those with trauma histories, may strengthen integrative care approaches and help bridge the gap between psychological and biomedical PCOS management.

## Figures and Tables

**Figure 1 behavsci-15-01551-f001:**
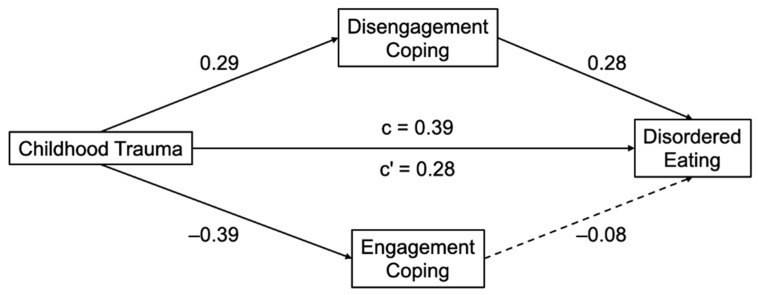
Mediation models linking childhood trauma to disordered eating via coping styles. Standardized regression coefficients shown. Solid lines represent significant paths (*p* < 0.05). Dashed lines represent non-significant paths. Models control for age, race, and education. Higher scores on the outcome indicate more disordered eating symptoms.

**Figure 2 behavsci-15-01551-f002:**
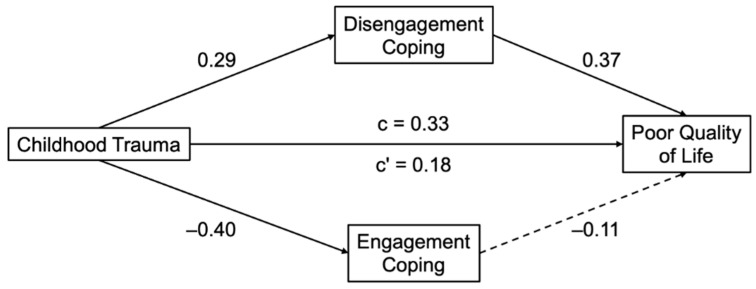
Mediation models linking childhood trauma to PCOS-related quality of life via coping styles. Standardized regression coefficients shown. Solid lines represent significant paths (*p* < 0.05). Dashed lines represent non-significant paths. Models control for age, race, and education. Higher scores on the outcome indicate poorer quality of life.

**Table 1 behavsci-15-01551-t001:** Descriptive statistics for key study variables.

Variable	M	SD	Range	α	Valid *n*
Childhood Trauma	52.5	20.5	25–108	0.89	150
Disordered Eating	3.3	1.4	0.3–5.7	0.93	149
PCOS-related Quality of Life	153.4	43.8	64–284	0.94	150
Engagement Coping	3.3	0.6	1.4–5.0	0.71	142
Disengagement Coping	3.3	0.7	1.8–5.0	0.76	142

## Data Availability

The data for this study are available at: https://osf.io/bqw9g/ (accessed on 30 July 2025).

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
