# Peer review of "Childhood Trauma, Coping Styles, and Mental Health in Polycystic Ovarian Syndrome"

_behavsci, 2025, doi:10.3390/bs15111551_

Round 1
Reviewer 1 Report
Comments and Suggestions for Authors
Thank you for the opportunity to review.
Summary: This manuscript examines whether coping styles mediate the relationship between childhood trauma and mental health outcomes (disordered eating and quality of life) among individuals with PCOS. The study is original, addressing a notable gap in the literature by exploring psychological mechanisms within a U.S.-based sample, and uses validated measures and appropriate analytic methods. Results indicate that disengagement coping partially mediates the association between childhood trauma and poorer outcomes. The manuscript is well-structured and grounded in relevant literature, though minor revisions could strengthen the paper.
General Comments:
The research question is timely and original, filling a clear gap in PCOS research. The hypotheses, methods, and analytic strategy are clearly stated. Results are presented transparently and supported by tables, figures, and appropriate statistical reporting. The discussion integrates findings with prior theory and research and acknowledges limitations.
Specific Comments:
Line 99
Please clarify how the Amazon gift card raffle was funded (e.g., departmental support, grant, personal funds).
Lines 113–117
The Coping Strategies Inventory description implies measurement of current coping. As someone not familiar with this specific scale (and unable to access it directly), it is unclear whether the measure reflects current or childhood coping, given the CTQ is retrospective. Please clarify.
Lines 122–127 The EDE-Q recoding is clearly described, but please discuss whether these modifications may affect comparability with prior studies. For the modified PCOSQ, construct validity may have shifted. Consider acknowledging this as a limitation.
Be mindful of wording that may imply causality. For instance:
- “explain” or “explained” (e.g., lines 17, 229, 320)
- “mediate” or “mediation” (e.g., lines 12, 198)
- “influence” (e.g., line 73)
Lines 237–238
Mediation effects (β = 0.08, β = 0.11) were statistically significant but modest. Consider explicitly noting this in the Discussion to avoid overinterpretation.
Line 292
Consider whether Reddit recruitment skewed participation toward younger participants. Was online recruitment chosen due to limited access to clinical recruitment pathways?
Lines 295–297
Consider expanding on the point that generic coping measures may not fully capture PCOS-specific stressors. Would the development of disease-specific coping instruments be a useful next step?
Line 306
Is racial homogeneity in the sample a limitation?
Multiple Testing / Type I Error
Several regression and mediation models were run. Please consider acknowledging the possibility of inflated Type I error risk, even if analyses were exploratory.
Reviewer 2 Report
Comments and Suggestions for Authors
The manuscript engages with an important and timely issue, and it is overall clearly written and thoughtfully structured. I commend the authors for their effort and valuable contribution. That said, I have several suggestions that may enhance both readability and conceptual clarity.
First, the section currently titled Introduction may benefit from revision. As there are no subsequent subheadings, the title appears somewhat redundant. Reframing it as Literature Review or incorporating additional subsections could improve organizational flow and assist readers in navigating the argument.
Second, in lines 31–35, the authors highlight that women frequently do not receive timely diagnoses or adequate clinical attention. This is a critical observation. However, linking this point more explicitly to the study’s aims and rationale would strengthen the argument. Moreover, while eating disorders and coping strategies are mentioned briefly toward the conclusion, engaging more extensively with this literature throughout the manuscript would help to better integrate and contextualize these themes.
With respect to data analysis, it would be useful to clarify whether formal hypothesis testing was conducted and whether the assumptions required for the primary analyses were met. Additionally, while age, race, and educational attainment are included in the analyses, these demographic variables should be introduced and justified earlier in the literature review. Positioning them within prior research would provide stronger theoretical justification for their inclusion.
Finally, I would like to thank the authors for openly sharing the anonymized data and for presenting a well-written and carefully prepared manuscript.
Reviewer 3 Report
Comments and Suggestions for Authors
Dear authors,
I find this article interesting because it addresses the relationship between childhood trauma and mental health problems in women with polycystic ovary syndrome.
My suggestions and comments on the article are as follows:
I believe the title is appropriate to indicate the topic addressed in the article and would facilitate searching for it in databases.
The abstract provides a comprehensive overview of the study.
The introduction addresses in sufficient detail the current issues regarding childhood trauma and mental health problems in women with polycystic ovary syndrome, although it should clarify why childhood trauma is specifically linked to this syndrome and whether the same result would not occur in other women who do not suffer from it.
The study hypotheses are clearly explained.
The methodology is explained in sufficient detail to allow replication of the study, although:
- The type of sampling used should be indicated.
- The implications of the respondents themselves deciding whether to formally diagnose themselves should be taken into account.
- It should be indicated whether the questionnaires used have been validated for the study population.
- The implications of this aspect should be analyzed in detail in the discussion: "Due to minor programming errors, the fifth anchor point on the seven-point scale was inadvertently omitted for the first 12 items, and Item 21 was rated on a four-point scale (not at all, slightly, moderately, and severely)."
The results are presented in detail with tables and figures for ease of understanding.
The discussion analyzes the results and establishes relationships with previous studies that address this topic. However, it should be addressed whether poorly addressed childhood trauma could be the actual cause of these patients' mental disorders, independently of polycystic ovary syndrome.
The exact meaning of this paragraph should be clarified: "Finally, the measure used to assess coping was previously validated but not tailored to the experiences of individuals living with PCOS."
The conclusions are consistent with the results obtained.
The references are appropriate for the topic investigated in this article.
Kind regards.
Round 2
Reviewer 3 Report
Comments and Suggestions for Authors
Dear authors,
I have no further comments or suggestions to make.
Kind regards.